

# The influence of mixing on stratospheric circulation changes in the 21st century

Eichinger Roland[1,2], Dietmüller Simone[2], Garny Hella[2,1], Šácha Petr[3,4], Birner Thomas[1],
Boenisch Harald[5], Pitari Giovanni[6], Visioni Daniele[7,#], Stenke Andrea[8], Rozanov Eugene[8,9],
Revell Laura[10,11], Plummer David A.[12], Jöckel Patrick[2], Oman Luke[13], Deushi Makoto[14],
Kinnison Douglas E.[15], Garcia Rolando[15], Morgenstern Olaf[16], Zeng Guang[16], Stone Kane Adam[17,18,*],
and Schofield Robyn[17,18]

[1]Ludwig Maximilians Universität, Meteorological Institute Munich, Munich, Germany
[2]Deutsches Zentrum für Luft- und Raumfahrt (DLR), Institut für Physik der Atmosphäre, Oberpfaffenhofen, Germany
[3]Faculty of Sciences, EPhysLab, Universidade de Vigo, Ourense, Spain
[4]Charles University Prague, Faculty of Mathematics and Physics, Department of Atmospheric Physics, Prague, Czech Republic
[5]Karlsruhe Institute of Technology (KIT), Insitute of Meteorology and Climate Reasearch, Karlsruhe, Germany
[6]Department of Physical and Chemical Sciences, Università dell'Aquila, L'Aquila, Italy
[7]Department of Physical and Chemical Sciences and center of Excellence CETEMPS, Università dell'Aquila, L'Aquila, Italy
[8]Institute for Atmospheric and Climate Science, ETH Zürich (ETHZ), Zürich, Switzerland
[9]Physikalisch-Meteorologisches Observatorium Davos and World Radiation Center, Davos, Switzerland
[10]Bodeker Scientific, Christchurch, New Zealand
[11]School of Physical and Chemical Sciences, University of Canterbury, Christchurch, New Zealand
[12]Environment and Climate Change Canada, Climate Research Division, Montréal, QC, Canada
[13]National Aeronautics and Space Administration Goddard Space Flight Center (NASA GSFC), Greenbelt, Maryland, USA
[14]Meteorological Research Institute (MRI), Tsukuba, Japan
[15]National Center for Atmospheric Research (NCAR), Boulder, Colorado, USA
[16]National Institute of Water and Atmospheric Research (NIWA), Wellington, New Zealand
[17]School of Earth Sciences, University of Melbourne, Melbourne, Australia
[18]ARC Centre of Excellence for Climate System Science, Sydney, Australia
[#]now at: Mechanical and Aerospace Engineering, Cornell University, Ithaca, New York, USA
[*]now at: Department of Earth Atmosphere and Planetary Science, Massachusetts Institute of Technology, Cambridge, Massachusetts, USA

*Correspondence to:* R. Eichinger (roland.eichinger@dlr.de)

**Abstract.**

Climate models consistently predict an acceleration of the Brewer-Dobson circulation (BDC) due to climate change in the 21st century. However, the strength of this acceleration varies considerably among individual models, which constitutes a notable source of uncertainty for future climate projections. To shed more light upon the magnitude of this uncertainty and on its causes, we analyze the stratospheric mean age of air (AoA) of 10 climate projection simulations from the Chemistry Climate Model Initiative phase 1 (CCMI-I), covering the period between 1960 and 2100. In agreement with previous multi-model studies, we find a large model spread in the magnitude of the AoA trend over the simulation period. Differences between future and past AoA are found to be predominantly due to differences in mixing (reduced aging by mixing and recirculation) rather





than differences in residual mean transport. We furthermore analyze the mixing efficiency, a measure of the relative strength of mixing for given residual mean transport, which was previously hypothesized to be a model constant. Here, the mixing efficiency is found to vary not only across models, but also over time in all models. Changes in mixing efficiency are shown to be closely related to changes in AoA and quantified to roughly contribute 10% to the long-term AoA decrease over the

21st century. Additionally, mixing efficiency variations are shown to considerably enhance model spread in AoA changes. To understand these mixing efficiency variations, we also present a consistent dynamical framework based on diffusive closure, which highlights the role of basic state potential vorticity gradients in controlling mixing efficiency and therefore aging by mixing.

**1  Introduction**

Air mostly enters the stratosphere through the tropical tropopause and then it ascends within the tropical pipe. Thereafter, it is transported poleward before descending to the extratropical lower stratosphere and back to the troposphere (Butchart, 2014). This stratospheric overturning cycle has been named the Brewer-Dobson circulation (BDC), referring to the early work of Dobson et al. (1929), Brewer (1949) and Dobson (1956), who first postulated this transport pattern on the basis of trace gas

observations. The structure and the strength of the BDC are notable sources of uncertainty for long-range climate projections as well as for short range weather forecasts (e.g. Hardiman and Haynes, 2008; Gerber et al., 2012; Kidston et al., 2015). The reasons for that are dynamical downward coupling (Baldwin and Dunkerton, 2001) and the BDC's influence on the distribution of radiatively active trace gases in the stratosphere. For example ozone and water vapour have an impact on Earth's radiative budget and thereby the surface temperatures (e.g. Solomon et al., 2010; Butchart, 2014). In addition, ozone protects humans

from excessive exposure to the harmful UV radiation (e.g. Thompson et al., 2011). Yet the representation of the strength, the structure and also the predicted future changes of the stratospheric overturning circulation differ vastly among today's state-of-the-art climate models (SPARC, 2010; Dietmüller et al., 2018), the same models that are applied to making predictions of the surface climate conditions across the 21st century.

Stratospheric mean age of air (AoA) is a commonly used diagnostic quantity for analyzing the BDC. It is defined as the

mean transport time of an air parcel from its entry into the stratosphere to any point therein (Hall and Plumb, 1994; Waugh and Hall, 2002) and thus reflects the transport patterns of the BDC. Also, this definition implies that AoA combines the effects of the slow overturning residual circulation as well as of the two-way mass exchange of air parcels, referred to as (eddy) mixing (Butchart, 2014). AoA is a common diagnostic in climate models, but it can also be derived from observations. Andrews et al. (2001); Engel et al. (2009) and Engel et al. (2017) made efforts to derive AoA from balloon-borne in situ measurements of

$CO_2$, $CH_4$, $N_2O$ and $SF_6$. These trace gases are suitable for such studies because they possess fairly long lifetimes and their tropospheric concentrations increased nearly linearly over the recent decades. A near global coverage of AoA observations was made possible for example through the work of Stiller et al. (2012) and Haenel et al. (2015), who derived it from satellite measurements of $SF_6$ and $CO_2$. The AoA observations of the recent decades and model simulations, however, do not tell the




same story. The time series of the observations presented in the studies by Engel et al. (2009); Ray et al. (2014); Engel et al. (2017) show a (non-significant) positive trend in the Northern Hemisphere (NH) across the recent decades, but most climate models show an AoA decrease over time. Also the trends in the (much shorter) satellite time series mostly do not coincide with the model results (Ploeger et al., 2015a). This discrepancy is still an ongoing debate and it has been addressed in numerous

studies (e.g. Garcia et al., 2011; Bönisch et al., 2011; Stiller et al., 2017; Lossow et al., 2018).

In the present study, we focus on the AoA differences between future and past simulated by 10 chemistry-climate models that participated in the Chemistry Climate Model Initiative phase 1 (CCMI-1; Morgenstern et al., 2017). We analyze the changes of stratospheric AoA in the 1960-2100 climate projection simulations between the two periods 1970-1990 and 2080-2100. It is well established that in climate change simulations models predict an acceleration of the BDC, which consequently leads to

younger stratospheric air (e.g. Rind et al., 1990; Butchart and Scaife, 2001; Garcia and Randel, 2008). Stratospheric transport is therefore sensitive to varying greenhouse gas concentrations, but also other constituents like ozone depleting substances (ODS) have been shown to play a considerable role in modulation of stratospheric transport (e.g. Oman et al., 2009; Oberländer-Hayn et al., 2015; Polvani et al., 2017, 2018). ODSs on the one hand, act as greenhouse gases themselves, and on the other hand, lead to the chemical destruction of stratospheric ozone. However, multi-model intercomparison studies have revealed that the

magnitude of the BDC acceleration until the end of the century varies strongly among the various state-of-the-art climate models (Butchart et al., 2006, 2010; Hardiman et al., 2014). Moreover, the mechanisms driving these changes are still not entirely clear.

To analyze the reasons for the AoA changes and their differences between various chemistry climate models (CCMs), we follow the approach of Birner and Bönisch (2011) who calculated the residual circulation transit times (RCTTs) by means of

backward trajectories. Garny et al. (2014) have then separated AoA into the two contributions of residual transport and aging by mixing. In several previous studies (e.g. Ploeger et al., 2015a; Dietmüller et al., 2017), it has been shown that this concept is well-suited for process-based model analyses. Here, the method allows us to conclude that a large fraction of the change in the stratospheric circulation is due to aging by mixing and residual transport only regionally plays the primary role. But these conclusions can prove fallacious, because also interactions between these two processes (Ploeger et al., 2015a) have to be

considered. While the changes that originate from residual circulation changes mainly depend on the strengthening of tropical upwelling (see e.g. Randel et al., 2008; Butchart et al., 2011; Butchart, 2014), the origin of changes in mixing are widely uncharted. We therefore calculate the mixing efficiency, an independent measure for the relative strength of mixing for given residual mean transport changes (ratio of mixing mass flux to net mass flux) (Garny et al., 2014), across the 21st century by means of a one-dimensional transport model of the stratosphere in the CCMI-1 model simulations. In the companion paper,

Dietmüller et al. (2018) have already shown that the mixing efficiency can explain most of the climatological AoA model spread. In the present study, we quantify the impact of mixing efficiency (relative mixing strength) differences in individual model simulations. Moreover, we show the influence of mixing efficiency variations on the model spread in AoA changes. To conclude, we also provide a theoretical explanation for the reasons of the relative mixing changes, based on the role of the ratio of wave dissipation to potential vorticity gradients in controlling the mixing properties.





## 2 Data and methods

### 2.1 CCM simulations

In this study, we analyze the model output of 10 state-of-the-art CCM simulations. All these simulations were conducted in the framework of the Chemistry Climate Model Initiative phase 1 (CCMI-1, Morgenstern et al., 2017). An overview on the model
5   simulations that are used for analysis in this study and some aspects of their model setup is given in Tab. 1. Additionally, the name of the atmospheric model component is provided to demonstrate similarities between some of the models. This sub-set of models has been chosen on the basis of availability of the required data for the analyses in this study (AoA and residual circulation velocities).

| Model | Reference(s) | Resolution | Model top | Atm. model |
|---|---|---|---|---|
| ACCESS | Morgenstern et al. (2009, 2013); Stone et al. (2016) | 2.5°x3.75°, L60 | 84 km | HadGEM3 GA2 |
| CMAM | Jonsson et al. (2004); Scinocca et al. (2008) | T47L71* | 0.0008 hPa | CCCma AGCM3 |
| EMAC-L47 | Jöckel et al. (2010, 2016) | T42L47 | 0.01 hPa | ECHAM5.3.02 |
| EMAC-L90 | Jöckel et al. (2010, 2016) | T42L90MA | 0.01 hPa | ECHAM5.3.02 |
| GEOSCCM | Molod et al. (2012, 2015); Oman et al. (2011, 2013) | 2.0°x2.5°, L72 | 0.015 hPa | GEOS-5 |
| MRI | Deushi and Shibata (2011) | TL159**, L80 | 0.01 hPa | MRI-AGCM3 |
|  | Yukimoto et al. (2011, 2012) |  |  |  |
| NIWA-UKCA | Morgenstern et al. (2009, 2013) | 2.5°x3.75°, L60 | 84 km | HadGEM3 GA2 |
| SOCOLv3 | Stenke et al. (2013); Revell et al. (2015) | T42L39 | 0.01 hPa | ECHAM5.4.00 |
| ULAQ | Pitari et al. (2014) | T21L126 | 0.04 hPa | ULAQ CCM |
| WACCM | Marsh et al. (2013); Solomon et al. (2015) | 1.9°x2.5°, L66 | 140 km | CAM4 |
|  | Garcia et al. (2017) |  |  |  |

*CMAM uses a T47 spectral resolution, but physics and chemistry are performed on a linear transform grid of around 3.8 degrees resolution.

**The AGCM component and the chemitstry-transport component of the MRI model have different resolutions (TL159 and T42, respectively), and each component is connected via a coupler. The atmospheric fields from the AGCM component are interpolated to T42 and used online in the chemistry-transport component which treats the AoA tracer.

**Table 1.** Overview of the CCMs and their setups of the CCMI-1 REF-C2 simulations. Also, the atmospheric model component of the CCMs is provided to demonstrate inter-model dependencies. For the spectral models, the horizontal resolution is provided as triangular truncation of the spectral domain, with T21≈5.56°x5.56°, T42≈2.5°x2.5° and TL159≈1.1°x1.1°. ACCESS: Australian Community Climate and Earth-System Simulator; CMAM: Canadian Middle Atmosphere Model; EMAC: ECHAM MESSy Atmospheric Chemistry; GEOSCCM: Goddard Earth Observing System Chemistry-Climate Model; MRI: Meteorological Research Institute; NIWA-UKCA: National Institute of Water & Atmospheric Research - United Kingdom Chemistry and Aerosols; SOCOLv3: modeling tools for studies of SOlar Climate Ozone Links, version 3; ULAQ(CCM): University of L'Aquila climate-chemistry model; WACCM: Whole Atmosphere Community Climate Model;



Note that ACCESS and NIWA-UKCA as well as EMAC-L47, EMAC-L90 and SOCOLv3 share the same atmospheric model component and that the two EMAC versions only differ in vertical resolution. ACCESS and NIWA-UKCA only differ by the facts that NIWA-UKCA is coupled to an ocean model and that the simulations are run on different platforms. The simulations we analyze are seamless simulations spanning the period 1960-2100, the so-called reference simulations REF-C2 (only r1i1p1 ensemble members). The simulations follow the WMO (2011) A1 scenario for ozone-depleting substances and the RCP 6.0 scenario (Meinshausen et al., 2011) for other greenhouse gases, tropospheric ozone ($O_3$) precursors, and aerosol and aerosol precursor emissions. For anthropogenic emissions, the recommendation was to use MACCity (Granier et al., 2011) until 2000, followed by RCP 6.0 emissions. Out of the models used in this study, MRI, NIWA-UKCA and WACCM have coupled an interactive ocean/sea ice module in these simulations for atmosphere-ocean interactions. In all other simulations, climate model fields (i.e., sea surface temperatures and sea ice concentrations) are imposed. A variety of different climate model data sets were used for this purpose (e.g. HadISST1 or HadGEM2 data, for details see table S1 in Morgenstern et al., 2017). More details on the simulation setups can be found in Morgenstern et al. (2017) and in the citations given in Tab. 1.

## 2.2 Analysis methods

The methodology of this study mostly follows the companion paper Dietmüller et al. (2018). A short description of the most important concepts used here is given in the following.

Stratospheric mean AoA is defined as the mean residence time of an air parcel in the stratosphere (Hall and Plumb, 1994; Waugh and Hall, 2002). In the CCMs, the AoA tracer is implemented as an inert tracer with a mixing ratio that linearly increases over time as lower boundary condition ("clock tracer" Hall and Plumb, 1994). In some models, this lower boundary condition is global, in others only in the tropics, in NIWA-UKCA and ACCESS for example, AoA is kept at 0 in the boundary layer. AoA is then calculated as the time lag between the local mixing ratio at a certain grid point and the current mixing ratio at a reference point. This reference point, however, varies between the models (e.g. boundary layer, tropopause, 100 hPa), which could lead to inconsistencies in the AoA calculation. To avoid this, we subtract the mean AoA value of the respective model's tropical tropopause from the actual AoA value at all grid points. This means that in our analyses, AoA=0 at the tropical tropopause for all models. We perform this calculation for each time step (with monthly values) separately for the entire time period 1960-2100. Therefore, the AoA trend exludes any changes in tropospheric transport times due to the fact that the tropopause rises over time (see Vallis et al., 2015; Oberländer-Hayn et al., 2016; Abalos et al., 2017).

The residual circulation transit time (RCTT) is the hypothetical age that air would have if it only followed the residual circulation, meaning that no processes such as eddy mixing or diffusion would come into play. These RCTTs are calculated using a concept described by Birner and Bönisch (2011), by calculating backward trajectories on the basis of the Transformed Eulerian Mean (TEM) meridional ($\overline{v}^*$) and vertical ($\overline{w}^*$) velocities (referred to as residual velocities, available in the CCMI-1 data base for the chosen models) with a standard fourth-order Runge-Kutta integration. The RCTT is then the time that these backward trajectories require to reach the tropopause from the respective starting point in the stratosphere. For more details see also Birner and Bönisch (2011) and Garny et al. (2014).



The RCTT differs from AoA because of (resolved and unresolved) mixing. In the stratosphere, this is due to the mixing of air between branches and the in-mixing of air from the mid-latitudes into the tropical pipe, which leads to recirculation of old air around the BDC branches. In global model studies, this effect has been named aging by mixing (A_mix) and is interpreted as the difference between AoA and RCTT (Garny et al., 2014; Ploeger et al., 2015a, b). However, it has to be kept in mind that the residual of AoA and RCTT does not only reflect this process alone, but actually includes resolved mixing as well as parameterized and numerical diffusion.

As a measure of the relative strength of mixing (independent of the residual circulation strength), we use the so-called mixing efficiency $\epsilon$ for analysis. $\epsilon$ is defined as the ratio of the mixing mass flux to the net residual mass flux between the tropics and the extratropics across the subtropical barrier. The net mass flux is the horizontal motion that is determined by mass continuity via vertical motion and corresponds to transport by $\overline{v}^*$ (Garny et al., 2014). $\epsilon$ can be derived by means of the tropical leaky pipe (TLP) model (Neu and Plumb, 1999). The TLP model is a one-dimensional transport model of the stratosphere that includes advection and horizontal mixing of air between the tropics and the extratropics. It assumes two columns of well-mixed air (a tropical and an extratropical column) and can be used to quantify the strength of mixing across the subtropical barrier. If we neglect vertical diffusion (see Dietmüller et al., 2017), we can formulate an analytical solution for tropical and mid-latitude AoA. According to the TLP model, tropical AoA ($AoA_T$) with altitude-dependent vertical velocity $w_T^*(z)$ can thus be described as

$$
\begin{aligned}
AoA_T(z) &= \int_{z_t}^{z} \frac{1}{w_T^*(z')} dz' + \epsilon \frac{\alpha+1}{\alpha} \left( \int_{z_t}^{z} \frac{1}{w_T^*(z')} dz' + H \left( \frac{1}{w_T^*(z)} - \frac{1}{w_T^*(z_t)} \right) \right) \\
&= RCTT(z) + \epsilon \frac{\alpha+1}{\alpha} \left( RCTT(z) + T_{corr}(z) \right).
\end{aligned}
\tag{1}
$$

Here, H denotes the scale height (7 km), $\alpha$ the ratio of tropical to extratropical mass approximated by

$$
\alpha = \frac{sin(lat(w^*(z)=0)) - sin(1 - lat(w^*(z)=0))}{2 \cdot sin(90) - sin(lat(w^*(z)=0))},
\tag{2}
$$

$z_t$ the height of the tropical tropopause and $T_{corr}(z) = H \left( \frac{1}{w_T^*(z)} - \frac{1}{w_T^*(z_t)} \right)$ the correction term for the altitude-dependency of the vertical residual velocity $w^*$. AoA thus depends on the advective vertical velocity $w_T^*$ (i.e. the residual velocity) and on the mixing efficiency $\epsilon$ (i.e. the amount of mixing between the tropics and the extratropics). Solving Eq. 1 for the mixing efficiency yields

$$
\epsilon = \frac{AoA_T(z) - RCTT(z)}{(RCTT(z) + T_{corr}(z)) \cdot \frac{\alpha+1}{\alpha}}.
\tag{3}
$$

Eq. 3 shows that $\epsilon$ is approximately proportional to the relative increase in AoA due to mixing. Note that according to the concept of the TLP model, $\epsilon$ has to be viewed as a parameter that counts for a given climate state in a certain model. Hence, it can vary between models and/or for different climate conditions. In the study by Garny et al. (2014), however, the authors nevertheless found a constant $\epsilon$ for three different climate states in one model. The tropical profiles provided for the TLP model are averaged over the latitudinal band of the models' individual vertically averaged turnaround latitudes (which are also time-dependent) and are interpolated to vertical coordinates relative to the tropopause height of each model. The mixing efficiency





is then obtained by a best fit for the altitude range from the tropopause to $32\,\mathrm{km}$ (details for the calculation of the mixing efficiency are given in Garny et al., 2014). According to the TLP formulation, aging by mixing (A_mix) is therefore a function of the mixing efficiency and of the residual circulation strength:

$$A\_mix = AoA - RCTT = \epsilon \cdot \frac{\alpha + 1}{\alpha} \cdot \left( RCTT(z) + T_{corr}(z) \right) \tag{4}$$

A_mix is proportional to the mixing efficiency, but indirectly proportional to the vertical velocity. A higher mixing efficiency is e.g. connected with more air parcels recirculating (see Garny et al., 2014), thereby increasing A_mix. But also, the vertical velocity controls the speed of the air parcels that recirculate. Thus, the mixing efficiency has been shown to be a useful diagnostic tool, as it does not depend on the speed of recirculation (e.g. Garny et al., 2014; Dietmüller et al., 2017).

## 3   Results

### 3.1   Changes in AoA and in its components

A multi-model comparison of stratospheric transport changes in the 21st century has been conducted before for the Stratosphere-Troposphere Processes and their Role in Climate (SPARC) report (SPARC, 2010). In that study, the authors showed stratospheric mean AoA of 10 chemistry-climate model simulations that took part in the CCMVal-2 (Chemistry-Climate Model

Validation, Eyring et al., 2013) project. To allow a direct comparison of the simulations that were analyzed in SPARC (2010) with the simulations we use here, Fig. 1 presents the same depiction of AoA differences between the 2090s and the 1990s ($\Delta$AoA) a) at $50\,\mathrm{hPa}$ and b) as difference between tropics and middle latitudes as in their figure 5.18.

The general structures of the AoA differences agree among the two model intercomparison projects. All models predict a decrease in mean AoA at $50\,\mathrm{hPa}$ and the smallest decrease in the tropics (Fig. 1a). A second minimum of the decrease is found

at $60^\circ$ N/S and the greatest decrease at $30^\circ$ N/S and/or at the poles. The AoA behaviour in these regions of maximum change across the 21st century is investigated in detail by Šácha et al. (2018). They found out that the trends there are related to the climatological AoA distribution, the upward shift of the pressure levels and the widening of the AoA isolines. In comparison with the CCMVal-2 models, the inter-model spread of the AoA difference is reduced in our results. However, it were the two UMUKCA (Unified Model/U. K. Chemistry Aerosol) models that lead to the large spread in the SPARC report and these

models are not part of our analysis. However, NIWA-UKCA and ACCESS are the direct successors to the UMUKCA models and these range rather in the lower field of AoA changes here. The ULAQ model has changed from a very large latitudinal $\Delta$AoA amplitude to a rather small one from CCMVal-2 to CCMI-1. Other models that appear in both studies do not show large changes. The EMAC model was not included in the SPARC report. With higher vertical resolution (47 to 90 levels in the vertical), the EMAC model tends to simulate larger $\Delta$AoA and thereby detaches from the bulk of the other model simulations.

In consistence with that, Revell et al. (2015) show that also in the SOCOLv3 model, AoA gets on average one year older when the model is run with 90 layers in the vertical (instead of 39) because of less vertical diffusion. All the statements above also count for panel b) with the tropical to middle latitude AoA differences with altitude (Fig. 1b). The spread is somewhat





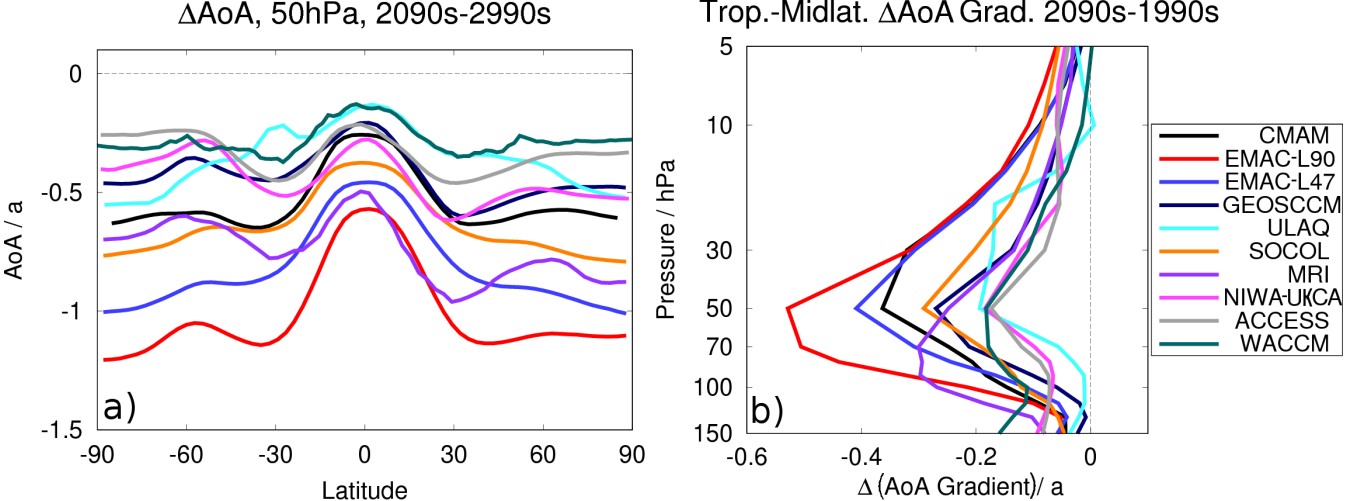

**Figure 1.** Mean AoA differences in the CCMI-1 REF-C2 simulations between the 2090s and the 1990s a) at 50 hPa with latitude and b) as gradient between tropical and middle latitudes with height. The depiction follows figure 5.18 of the SPARC CCMVal-2 report (SPARC, 2010).

smaller (mainly because of the UMUKCA models), ULAQ changes strongly and the other models show a similar behaviour in CCMVal-2 and in CCMI-1.

To visualize AoA and its trends of the model simulations and their inter-model differences, we present the annual mean AoA data of the 10 CCMI-1 model simulations in Fig. 2. Moreover, the piecewise linear regression of AoA for the periods 1960-2000

and 2000-2100 is presented. These two periods were chosen because the year 2000 marks a change in stratospheric dynamics, which is due to the reversal in sign of ODS and ozone trends as a consequence of the Montreal Protocol (see Morgenstern et al., 2018; Polvani et al., 2018). We chose the 30 hPa pressure level and an average between 30°N and 50°N, because the observation-based data from Engel et al. (2009, 2017) are from that region too. These data and their linear regression are included in the figure as well, they show a non-significant positive AoA trend of 0.15±0.18 years/decade.

To begin with, the absolute AoA values differ strongly among the models. For the first three decades of the simulations (i.e. the mean from 1960-1990) the values range between 5.52 years in the MRI model and 3.18 years in the ACCESS model simulation. This topic had already been discussed for example in SPARC (2010) and in Dietmüller et al. (2018). Analysing the hindcast simulations of the CCMVal-2 and the CCMI-1 projects, Dietmüller et al. (2018) showed that it is mainly the mixing rather than the residual circulation that causes the large AoA spread and that this is likely linked to the different resolutions of

the model simulations.

All the model simulations show a clear negative trend over time across the first as well as across the second period. The in situ measurement-based observations (Engel et al., 2009, 2017), in contrast, display a positive, but not significant, trend for the recent couple of decades. A similar behaviour as in the in situ measurements can also be found in satellite-based




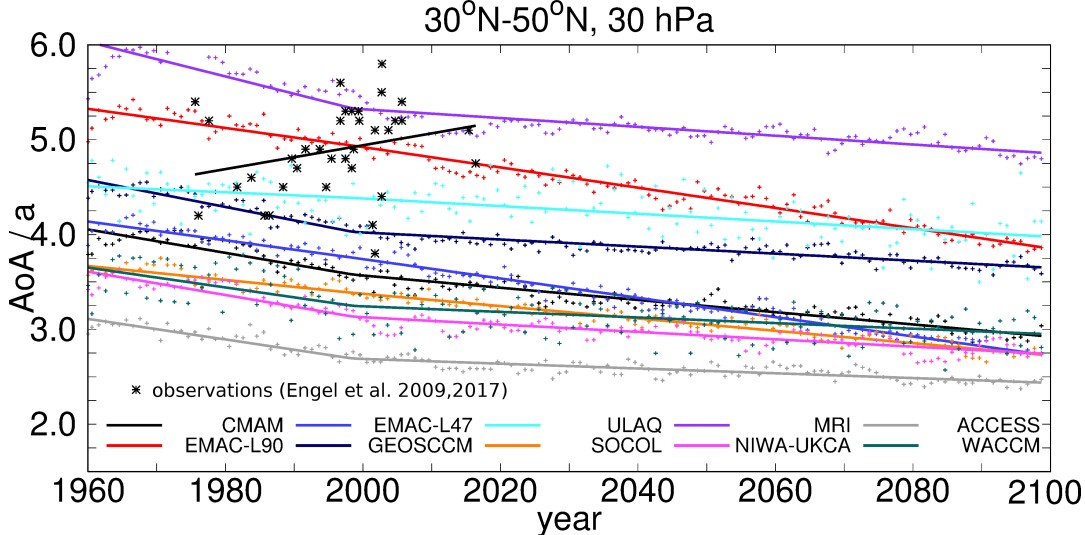

**Figure 2.** Monthly mean AoA of the REF-C2 CCMI-1 model simulations (dots) and their linear trends at 30 hPa averaged over $30°N-50°N$ as well as AoA derived from in situ measurements by Engel et al. (2009, 2017) (stars). Note that the observational AoA data is relative to ground level, while the model data has been processed to be relative to the tropopause (see main text).

observations for example in Haenel et al. (2015), although for a shorter time series (2002-2012). A number of studies have investigated this discrepancy (see e.g. Engel et al., 2009; Garcia et al., 2011; Ploeger et al., 2015a) and many reasons have been discussed to resolve this contradiction (e.g. effect of mixing in models, sparse sampling of observational data, differences in changes between deep and shallow BDC branches), but a viable explanation is still missing. In the present study, however,

we do not discuss this issue any further, but rather focus on the analysis of the negative AoA trends in the model simulations. The models may agree in predicting a decrease in stratospheric AoA, but they do show large differences in the strength of this trend. Also, the trends between the two periods (1960-2000 and 2000-2100) differ. Most models show a stronger trend between 1960 and 2000, only the two EMAC simulations have a stronger trend in the second period and in SOCOLv3 the trend almost remains constant. Note, however, that EMAC has a negative bias in ODSs, because the replacement products were

not taken into account as F11 or F12 equivalents. This can possibly explain why the AoA trend is weaker between 1960 and 2000. Several studies (e.g. Austin and Li, 2006; Oberländer-Hayn et al., 2015; Oman et al., 2009; Polvani et al., 2017, 2018; Morgenstern et al., 2018) have recently pointed out the importance of the role of ODSs (as both, radiatively active greenhouse gases and chemically active gases controlling ozone depletion and recovery) for the trends in stratospheric dynamics. During the period between 1960 and 2000, ODSs increase and thus act in concert with GHGs to cause an AoA decrease. Thereafter,

however, ODSs decrease over time, which means that with respect to AoA trends, the ODS trend works against the trend in the continuously rising GHGs. This can possibly explain the weakening in the trend in most models in the second period in Fig. 2. Due to this change in dynamical properties around the year 2000, an analysis of the trends across the entire period from



1960 to 2100 can not be conducted without mixing-up various dynamical effects. The first period is relatively short for robust analyses of mixing trends and in the second period, two mechanisms work against each other, so that in most models, the trends are rather small. In the following, we therefore analyze the differences between the periods 1970-1990 and 2080-2100. This separates the GHG effect from the ODS effect. At the end of the 21st century, ODS mixing ratios have declined to similar values as between 1970 and 1990 in the simulations. We do not start our investigation at 1960 due to the calculation method of RCTTs and $\epsilon$ which are available only from 1970 onwards.

| | AoA / a | | | $\epsilon$ / $10^{-1}$ | | | trop. upw. / $10^9$kg/s | | |
| **Model** | 1970 | 2100 | $\Delta$ | 1970 | 2100 | $\Delta$ | 1970 | 2100 | $\Delta$ |
|---|---|---|---|---|---|---|---|---|---|
| ACCESS | 2.35 | 1.97 | -0.38 | 4.00 | 4.03 | 0.03 | 7.92 | 8.97 | 1.05 |
| CMAM | 2.73 | 2.19 | -0.54 | 4.02 | 3.92 | -0.09 | 8.05 | 9.68 | 1.63 |
| EMAC-L47 | 2.69 | 1.96 | -0.72 | 5.05 | 4.27 | -0.78 | 8.92 | 11.38 | 2.45 |
| EMAC-L90 | 3.43 | 2.60 | -0.82 | 6.46 | 5.42 | -1.03 | 7.84 | 10.13 | 2.29 |
| GEOSCCM | 3.17 | 2.68 | -0.49 | 4.53 | 4.14 | -0.38 | 6.90 | 7.73 | 0.83 |
| MRI | 4.22 | 3.49 | -0.73 | 9.04 | 8.35 | -0.69 | 7.80 | 9.02 | 1.21 |
| NIWA-UKCA | 2.71 | 2.20 | -0.51 | 4.60 | 4.65 | 0.04 | 9.81 | 12.04 | 2.22 |
| SOCOLv3 | 2.47 | 1.94 | -0.53 | 4.38 | 3.80 | -0.59 | 8.92 | 1.06 | 1.70 |
| ULAQ | 2.82 | 2.56 | -0.25 | 5.64 | 5.98 | 0.34 | 7.34 | 8.41 | 1.07 |
| WACCM | 2.62 | 2.25 | -0.39 | 3.79 | 3.60 | -0.18 | 7.97 | 8.94 | 0.96 |

**Table 2.** Climatologies and differences of AoA (averaged over 100-10hPa and 90°S-90°N), the mixing efficiency $\epsilon$ and tropical upwelling at 70 hPa. 1970 means averaged over 1970-1990 and 2100 means averaged over 1980-2100. $\Delta$ means the difference between the two latter (climate states 2100 minus 1970). Note that rounding can lead to seemingly wrong $\Delta$ values here.

Tab. 2 provides an overview over mean AoA, $\epsilon$ and tropical upwelling (at 70 hPa) averaged between 1970 and 1990 (denoted as 1970) and between 2080 and 2100 (denoted as 2100) and their differences ($\Delta$) for the ten CCMI-1 REF-C2 model simulations. For consistency with $\epsilon$ and tropical upwelling (which are global values), we here averaged AoA globally (90°S-90°N) and over 10-100 hPa, which means that these AoA values are not comparable with those in Fig. 2. These values will be used to explain or confirm some of the results throughout the paper.

Fig. 3 shows the linear AoA changes along with the ratio of the RCTT change to the AoA change ($\Delta$RCTT/$\Delta$AoA) for model intercomparison of the zonal mean structure of the differences. Moreover, the contour lines show the climatologies of the 1970-1990 period of the simulations for AoA and for the RCTT/AoA ratio, respectively. The $\Delta$RCTT/$\Delta$AoA ratio provides the possibility to linearly separate $\Delta$AoA into its contributions from residual transport and from aging by mixing (A_mix). Hence, values between 0.5 and 1 signify a domination of residual transport changes (reddish), while values between 0 and 0.5 mean that the $\Delta$AoA is controlled by A_mix variations (blueish). Values above 1 mean that the A_mix difference is positive and values below 0 mean that $\Delta$ RCTT is positive (assuming a negative $\Delta$AoA).





**Figure 3.** ΔAoA (colummns a and c) and ΔRCTT/ΔAoA (columns b and d) between the periods 1970-1990 and 2080-2100 for the CCMI-1 REF-C2 simulations in colours. The contour lines show the respective 20 year climatologies from 1970-1990. Stippled regions mark where the significance of the difference is below the threshold of 95%.



In most models, a quite similar ΔAoA pattern can be seen. Relatively small ΔAoA dominates the tropical pipe where young air is prevalent and the differences increase with higher altitudes and latitudes. Several model simulations (ACCESS, CMAM, MRI, NIWA-UKCA), however, show their maximum ΔAoA in the lower stratospheric middle latitudes. This feature is connected with the climatological AoA gradient, the upward shift of the circulation and the decrease in RCTT and A_mix in Šácha et al. (2018). As already indicated above, the two EMAC simulations show the largest changes throughout the stratosphere and the ULAQ model has the weakest changes. Moreover, as the only model, ULAQ shows a pronounced hemispheric asymmetry with stronger differences in the SH. These general ΔAoA patterns had also been found in the multi-model study by Butchart et al. (2010) who analyzed 11 21st century model simulations that were performed for WMO (2006). Most models also agree in the ΔRCTT/ΔAoA pattern. Strong A_mix-dominated ΔAoA can be seen in the middle latitude lower stratosphere and the residual circulation dominates ΔAoA mainly in the tropical pipe as well as in the downwelling branches in the high latitudes. Some model simulations hardly show any reddish colours (EMACL47, EMACL90, GEOSCCM, MRI), which means that the differences of residual transport accounts for less than 50% of ΔAoA throughout the stratosphere, even in the upwelling and downwelling branches. Other models (CMAM, ACCESS, SOCOLv3, WACCM) show distinct red upwelling and downwelling branches. This indicates that ΔAoA is not influenced much by ΔA_mix in these particular regions. The NIWA-UKCA model shows a clear separation between aging by mixing and residual circulation dominated regions, and in the ULAQ model, the ΔA_mix is slightly positive in most parts of the stratosphere. Altogether, these results suggest that changes in aging by mixing have a major impact on ΔAoA. This brings us to further analyze the A_mix changes as well as the mixing efficiency and to discuss possible reasons for their variations.

### 3.2 Coherences between the components

Fig. 3 suggests that most of the ΔAoA is determined by aging by mixing changes. However, A_mix itself depends on RCTT (see Eq. 4) and is therefore not an independent measure for separation of the processes. AoA changes are commonly attributed to changes in the residual circulation (e.g. Austin and Li, 2006). In the climatologies, the tropical upwelling (as calculated via the vstar integral at the turn-around latitudes, see e.g. Okamoto et al., 2011) in the 70 hPa pressure level has been shown to be a good measure for the strength of the residual circulation throughout the stratosphere (see Dietmüller et al., 2018). To see if this relationship holds true for ΔAoA in our set of the CCMI model simulations, we present the inter-model correlations between ΔAoA and ΔRCTT as well as between the ΔAoA and the differences in tropical upwelling in 70 hPa in Fig. 4. These correlations are calculated across the 10 model simulations for each grid point separately. Note that for this, the data of each model was interpolated to the resolution of the grid of the model with the highest horizontal resolution.

A high correlation between ΔRCTT and ΔAoA (Fig. 4a) can only be seen in the middle to high latitudes between around 70 and 10 hPa and low or no correlations in the other regions. Particularly in the upwelling region in the tropics, this is rather surprising since AoA should be controlled by the upwelling speed there. However, this connection apparently is not local. The correlations between ΔAoA and Δtropical upwelling (Fig. 4b) are high in the entire stratosphere above around 70-50 hPa. This reflects a clear connection between tropical upwelling and AoA, particularly in the deep branch. A stronger tropical upwelling generates a faster circulation and hence a decrease in AoA. The strengthening of tropical upwelling can be explained by an



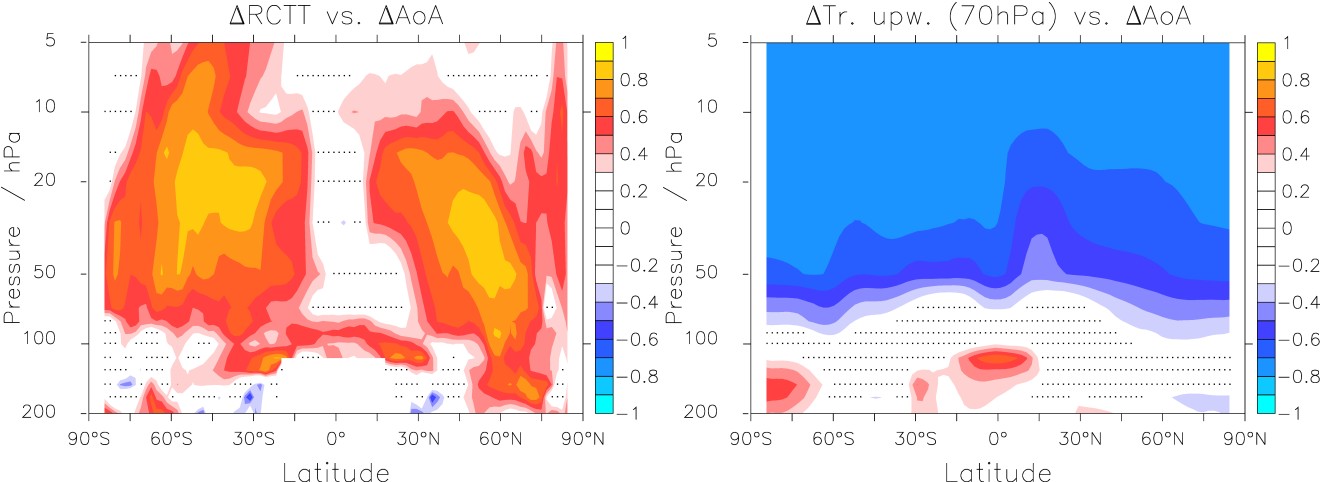

**Figure 4.** Inter-model correlations a) between $\Delta$AoA and $\Delta$RCTT and b) between $\Delta$AoA and $\Delta$tropical upwelling at 70 hPa. Stippled regions show where the significance level of the correlation is below the threshold of 5%.

enhancement of the subtropical jet streams following from upper tropospheric warming and lower stratospheric cooling (see e.g. Lorenz and DeWeaver, 2007; Randel et al., 2008; Butchart, 2014) and is linked with the upward shift of the tropopause by Vallis et al. (2015); Oberländer-Hayn et al. (2016). The rather patchy picture of the $\Delta$RCTT vs. $\Delta$AoA correlations suggests that the link between the residual circulation and AoA can not be expected to be local, but is rather of remote nature. It is

mainly the ascent of air in the tropics that determines the influence that changes of the residual circulation have on $\Delta$AoA. Particularly the upwelling and downwelling regions do not correlate well in Fig. 4a, although $\Delta$AoA is clearly dominated by the changes in transport in those regions (see Fig. 3).

    Dietmüller et al. (2018) have found out that the climatological inter-model AoA spread in the CCMVal-2/CCMI-1 hindcast simulations can mostly be explained with model differences in mixing efficiency and only to some extent with differences in

the residual circulation. Hence, we now lay our focus on the model trends (and differences between the two periods) in mixing efficiency $\epsilon$ and on the processes that are responsible for its changes. $\epsilon$ provides information on the models' relative strengths in mixing at the tropical barriers as a vertically integrated measure (see Sect. 2.2 for calculation method). Fig. 5 shows the time series of the mixing efficiency of each model for the REF-C2 simulation period and as in Fig. 2, the piecewise linear regressions for the time before and after the year 2000 are included. Note that due to the calculation of the 10 year moving

average of $\epsilon$, the first ten years can not be assessed here. For a more quantitative view, the $\Delta\epsilon$ and the climatological $\epsilon$ of the two simulation periods are also presented in Tab. 2.

    The climatological $\epsilon$ of the first 20 REF-C2 simulation years show similar values as the REF-C1 (CCMI-1 reference hindcast simulations from 1960-2011, for details see Morgenstern et al., 2017) values that were shown in Dietmüller et al. (2018). In this period, the main differences between the REF-C1 and the REF-C2 simulations are the sea surface temperatures and the

sea ice distribution. Only MRI shows a somewhat higher mixing efficiency in the REF-C2 simulation ($\sim$ 50%). The multi-





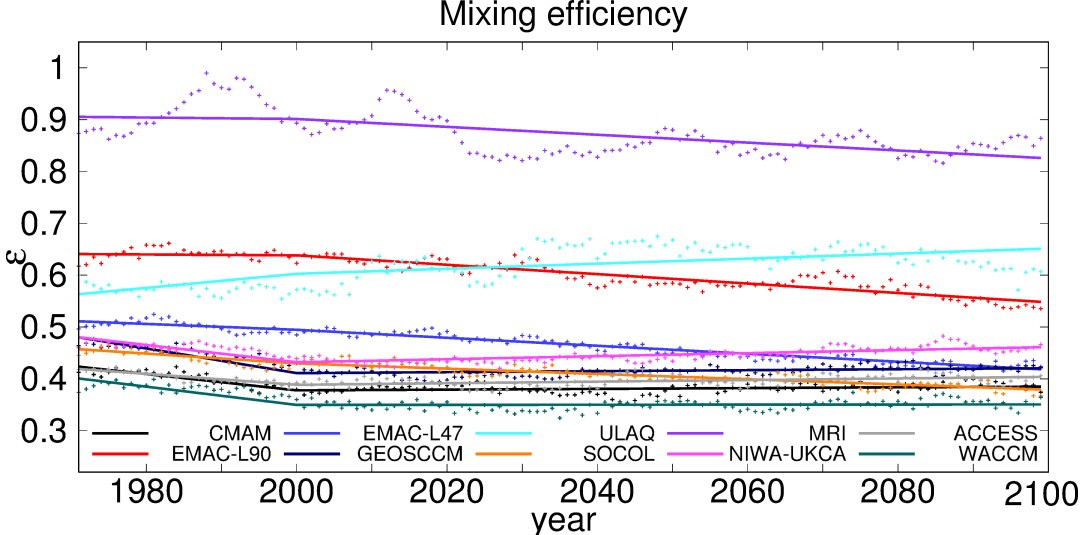

**Figure 5.** Mixing efficiencies $\epsilon$ of the REF-C2 CCMI-1 model simulations and their piecewise linear regression for the periods 1970-2000 and 2000-2100.

model mean of this climatological $\epsilon$ is 0.52±31%. This agrees well with the value (0.58±32%) that was found for the REF-C1 simulations (for $\epsilon$ calculated at the turnaround latitudes) by Dietmüller et al. (2018). Across the whole simulation period, the mixing efficiency decreases over time in most models. However, the sign of the trend varies a lot between the two periods. For example in MRI and in EMAC-L90, $\epsilon$ first increases and then decreases and in GEOSCCM and CMAM this is vice versa. For

the reasons we mentioned above (counteracting influence of GHGs and ODSs), we now again discuss the differences between the start and the end of the simulations to filter the direct effect of ODSs on stratospheric dynamics. The $\epsilon$ values for the periods in question as well as their differences are provided in Tab. 2. The ACCESS and the NIWA-UKCA model (the two models with the HadGEM atmospheric model component) show a slight positive $\Delta\epsilon$ and the only model with a strong positive $\Delta\epsilon$ is ULAQ. The fact that the mixing efficiency is not constant over time means that the absolute mixing strength does not change

proportionally to the residual circulation. This is in contrast to the results of Garny et al. (2014), who found a nearly constant $\epsilon$ when comparing three GCM simulations with varying climate states. However, Garny et al. (2014) also make clear that it would not be surprising if $\epsilon$ varied over the course of climate change simulations, because also the properties of the mixing barriers can change over time. These are diagnosed using the potential vorticity gradient, which mainly depends on the zonal winds, but see Sect. 3.4 for an analysis of the mechanisms of the mixing changes.

In consequence, we next investigate if, apart from their connection to residual circulation changes, $\Delta$AoA is also linked to variations in mixing efficiency. For this, Fig. 6 shows the inter-model correlations (correlations across the 10 model simulations) between the local $\Delta$AoA and the models' differences in mixing efficiency.




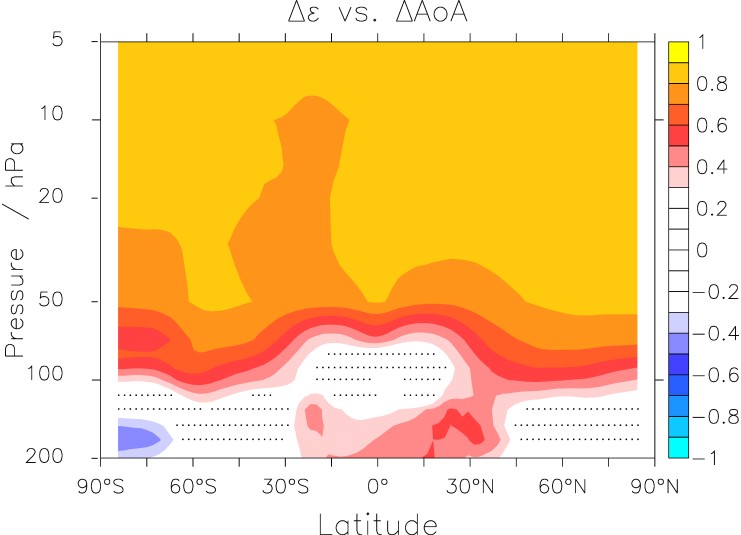

**Figure 6.** Inter-model correlations between $\Delta$AoA and $\Delta\epsilon$. Stippled regions show where the significance level of the correlation is below the threshold of 5%.

Fig. 6 shows a clear link between $\Delta$AoA and $\Delta\epsilon$ above around 100-70 hPa, with correlation coefficients mostly ranging between 0.8 and 0.9. This reflects that a decline in mixing efficiency leads to a decrease in AoA, because there is less recirculation of air around the BDC. Only in the region around the Antarctic polar vortex, the correlation is generally somewhat weaker. This may be due to changes in the polar vortex strength or position in the various models, which can not be reflected in the (sub)tropical measure $\epsilon$.

A_mix is connected with the mixing efficiency, but it also depends on the residual circulation. A higher/lower mixing efficiency causes more/less recirculation, which increases/decreases aging by mixing, and the RCTT also controls the transit times of the air parcels that recirculate. Overall, we can again conclude that $\Delta$AoA in climate models is connected with both, the changes in mixing and in residual transport. Most of the $\Delta$AoA that is connected with residual transport can be attributed to tropical upwelling changes, at least for the deep BDC branch. This is caused by a climate change induced strengthening and/or upward shift of the subtropical jet streams (see e.g. Randel et al., 2008; Shepherd and McLandress, 2011; Butchart, 2014; Oberländer-Hayn et al., 2016). The differences in mixing between future and past climate, however, have not been analyzed in detail before. Therefore, we further investigate the overall effect of mixing on the BDC changes in the remainder of the paper.

### 3.3 Impact of the mixing changes

To quantify the impact that changes in mixing efficiency have on $\Delta$AoA in the simulations, we again apply the TLP model, now by using $\overline{w}^*$, the tropopause height and a given $\epsilon$ to calculate the RCTTs and AoA (see Eq. 1). We then compare the AoA and RCTT climatologies of the two periods 1970-1990 (1970) and 2080-2100 (2100). Fig. 7 shows the tropical AoA





and RCTT profiles (calculated with the TLP model) of these climatologies exemplarily for the EMAC-L90 model simulation. Moreover, two hypothetical AoA profiles (AoA' and AoA*) are included in the figure (both for 2100). AoA'(2100) (gray line in Fig. 7) displays the AoA climatology for 2100 with the A_mix fixed to the value of 1970, it is calculated by $AoA'(2100) = AoA(1970) − (RCTT(1970) − RCTT(2100))$. The difference between AoA(1970) and AoA'(2100) then yields the AoA

5   change that is caused by the RCTT change only (without a change in A_mix). The AoA*(2100) (black line in Fig. 7) profile displays the AoA climatology of the 2100 climate state by keeping the mixing efficiency constant at the value of 1970. We calculate this quantity by using $\epsilon(1970)$ in Eq. 1 for the $AoA_T(2100)$ calculation, thus it represents how much influence the change in $\epsilon$ has on $\Delta AoA$ (in contrast to the change in residual circulation only).

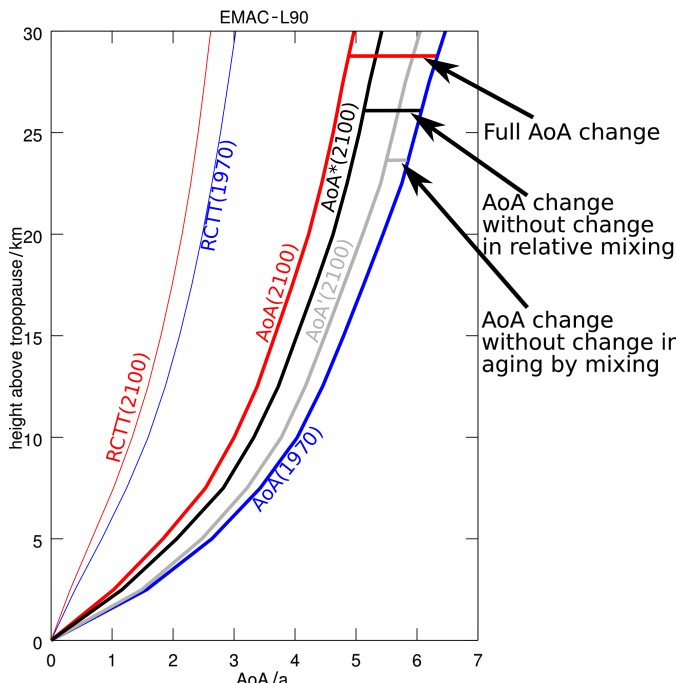

**Figure 7.** AoA and RCTT of the EMAC-L90 simulation for the two periods 1970-1990 and 2080-2100. Also included are a hypothetical AoA* that displays the AoA at the end of the simulation but with the mixing efficiency of the beginning of the simulation and AoA', a hypothetical AoA that represents AoA at the end of the simulation with the A_mix of the beginning (see main text for details). The difference between AoA and RCTT resembles the influence of mixing. The difference between AoA and AoA* (both 2080-2100) resembles the influence of the mixing efficiency changes on $\Delta AoA$ and AoA' represents the subdivision of the AoA difference between the two climate states into the influences of A_mix and RCTT changes.

First, the RCTT and AoA climatologies of the EMAC-L90 simulation of both climate states show that RCTT explains about

10   one third of the AoA values. The difference (about two thirds) is caused by aging by mixing difference (see Fig. 7). This ratio had already been found in Dietmüller et al. (2018) for the CCMVal-2 and CCMI-1 hindcast simulations. AoA' reveals that for





EMAC-L90 this ratio roughly also counts for the differences between the periods. The AoA difference between 2100 and 1970 is subdivided by AoA' into about one third the share of RCTT difference and two thirds of aging by mixing. This ratio can also be estimated from Fig. 3 for the EMAC-L90 simulation. However, as already stated above, the A_mix change also includes some RCTT change (see Eq. 4), which implies that this is not a clear separation of the mechanisms.

The impact of the mixing efficiency on $\Delta$AoA can be seen through AoA*. Or the other way round, the difference between AoA*(2100) and AoA(2100) reflects the impact of $\Delta\epsilon$ on AoA. In the given example of the EMAC-L90 model, the fractional impact of the mixing efficiency change on the AoA difference is 29±4% calculated between 2.5 and 25 km above the tropopause. This value varies considerably among the models. Tab. 3 provides an overview of the quantity for the ten models.

| Model | ACCESS | CMAM | EMAC-L47 | EMAC-L90 | GEOSCCM |
|-------|--------|------|----------|----------|---------|
|       | -3.5±0.5% | 10±1% | 24±2% | 29±4% | 17±2% |
| Model | MRI | NIWA-UKCA | SOCOLv3 | ULAQ | WACCM |
|       | 23±2% | -5.5±1.5% | 23.5±1.5% | -29±10% | 11.5±1.5% |

**Table 3.** Contribution of the relative change in mixing (i.e. the mixing efficiency changes) to the overall $\Delta$AoA between the periods 2080-2100 and 1970-1990 calculated between 2.5 and 25 km above the tropopause.

With less than 5.5% and 3.5%, NIWA-UKCA and ACCESS have the lowest contribution of $\Delta\epsilon$ on the AoA change and
with up to 29%, ULAQ and EMAC-L90 have the largest. But in the case of NIWA-UKCA, ACCESS and ULAQ, the mixing efficiency increases (see Tab. 2), which means it leads to an increase in AoA, and not to a decrease as in the other models. The negative $\Delta$RCTT therefore accounts for more than the entire negative $\Delta$AoA to compensate the effect of the $\epsilon$ change. The other models have a contribution of $\Delta\epsilon$ on $\Delta$AoA between 10 and 29%, the multi-model mean is 10.4% ($\sigma = 18.3\%$). It is reasonable that a large $\Delta\epsilon$ leads to a high percentage of the $\epsilon$-share on $\Delta$AoA. The correlation coefficient between the
two values is 0.83. This makes clear that the change in mixing efficiency does have a considerable impact on $\Delta$AoA, as it could already be assumed from the high inter-model correlations between the two quantities (Fig. 6). Due to the large model spread in $\Delta\epsilon$, however, this impact bears large uncertainties. Next, we quantify the impact of $\Delta\epsilon$ on the model spread in $\Delta$AoA.

To analyze the share of $\Delta\epsilon$ on the AoA model spread, we again use the TLP model-calculated values of $\epsilon$, $\alpha$, $T_{corr}$, and
$RCTT_T$. These quantities are taken to calculate tropical AoA by

$$AoA_T = RCTT_T + \epsilon \cdot \frac{\alpha+1}{\alpha} \cdot (RCTT_T + T_{corr}), \qquad (5)$$

as an average between 100 and 10 hPa. As above, this calculation is performed for each model for the two periods, respectively, so that $\Delta AoA_T = AoA_T(2100) - AoA_T(1970)$. Using $\epsilon(1970)$ in the $AoA_T(2100)$ calculation provides an $AoA_T^*(2100)$
and thus $\Delta AoA_T^* = AoA_T^*(2100) - AoA_T(1970)$ provides the AoA difference without any changes in $\epsilon$. This provides infor-



mation about the influence of the mixing efficiency on the spread in $\Delta$AoA. Fig. 8 displays the model distribution of $\Delta AoA$ with (blue) and without (red) a changing mixing efficiency.

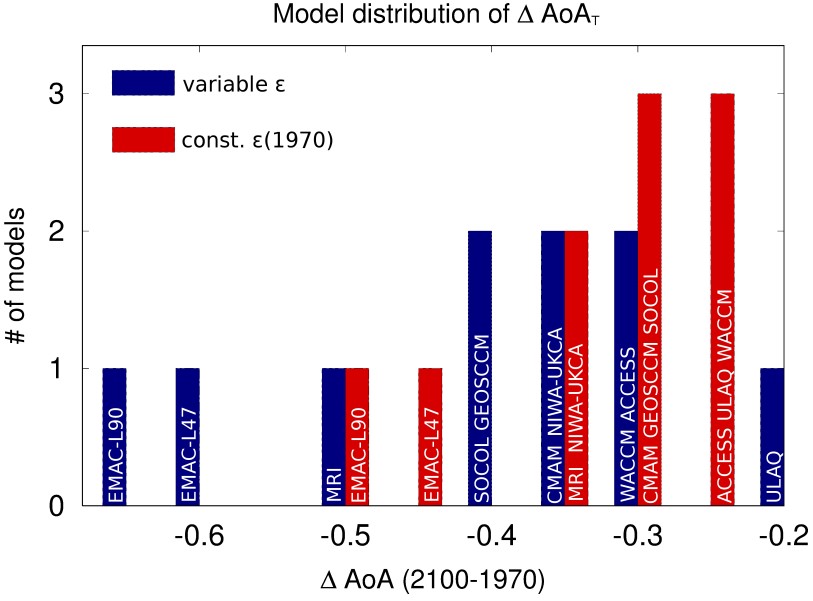

**Figure 8.** Model distribution of the tropical AoA difference (averaged from 100 to 10 hPa) between the two periods 1970-1990 and 2080-2100 calculated after Eq. 5 with a variable $\epsilon$ (blue) and with a constant $\epsilon(1970)$ (red).

Fig. 8 clearly shows that when $\epsilon$ is kept constant, first, $\Delta$AoA generally decreases (in absolute values) and second, the model range of $\Delta$AoA is considerably reduced (from 0.35 to 0.22). This reflects that the mixing efficiency changes lead to strong variations in AoA. When $\epsilon$ is held constant, for models with negative $\Delta\epsilon$, $\Delta$AoA is reduced (in absolute values) (see e.g. EMAC) and $\Delta$AoA is enhanced for models with positive $\Delta\epsilon$ (see e.g. ULAQ). However, it is not strictly coincidental that the model range decreases here (because being the model with the largest $\Delta$AoA, EMAC-L90 has a negative $\Delta\epsilon$ and ULAQ, the model with the lowest $\Delta$AoA has a positive $\Delta\epsilon$), because one can argue that a large/small (negative) $\Delta\epsilon$ actually causes a large/small $\Delta$AoA.25%-33% From this analysis, we can conclude that $|\Delta$AoA$|$ generally increases through the changes in mixing efficiency (by 10.4% as multi-model mean) and that $\Delta\epsilon$ leads to a larger model spread in the AoA changes ($\Delta\epsilon$ increases the $\Delta$AoA model range by about 37%).

Now, the question remains what the reasons for the changes in relative mixing strength are, or why $\epsilon$ changes in the course of climate change simulations. In the next section, we will explore an explanation for this by analyzing various dynamical fields.

### 3.4   On the mechanism of mixing changes

The relation of the residual circulation and mixing determines the mixing efficiency and changes therein. To study possible dynamical reasons for the changes in the mixing efficiency, the relation of wave driving, the residual circulation and mixing is



analyzed in the following based on the transformed Eulerian mean (TEM) momentum equation. This analysis is similar to that presented in Garny et al. (2014), but here we use the quasi-geostrophic (QG) formulations on pressure levels (on which the CCMI data are available). We present and analyze multi-model means (MMMs) diagnostics to provide a general picture for the entire subset of CCMI-1 model simulations. Unless otherwise stated in the text, the individual models qualitatively agree fairly

well in these diagnostics. The fields of the individual models can be found in the Supplement. Fig. 9 shows the differences between the two periods (1970-1990 and 2080-2100) overlaid with the MMM climatologies of the first period of zonal winds, Eliassen-Palm flux divergence (EPfd), $v^*$, the meridional potential vorticity gradient ($\partial PV/\partial y$), as well as $K_{yy}$ and $K_{yy}/|v^*|$. The calculation of $K_{yy}$ and the meanings of these variables will be explained below. Due to data availability, not all models could be included in these MMMs. For SOCOL and ULAQ, the EPfd fields were not available, hence, for consistency, all

MMMs base on the remaining eight other models only.

The mechanism of the increase in residual circulation in the lower stratosphere is well understood (e.g. Shepherd and McLandress, 2011). It follows the increase in upper tropospheric temperatures that leads to an upward shift and increase in the zonal mean winds. This can be seen in Fig. 9a, which shows the MMM differences of the zonal mean wind between the two time periods. Subsequently, the critical layers that allow for wave propagation shift to higher levels. This becomes evident from

Fig. 9b. There, a region of strongly negative $\Delta$EPfd (enhanced wave dissipation) can be seen in the MMM between about 100 and 50 hPa (with maxima around 70 hPa) and positive differences (less wave dissipation) can be found between around 200 and 150 hPa. The enhanced wave dissipation in the lower stratosphere leads to amplified poleward residual transport, which reflects in $\Delta$v$^*$ (Fig. 9c).

However, as shown in the sections above, transport changes in a future climate are not only due to this increase in residual

transport, but also due to changes in eddy mixing. The EPfd ($\nabla \cdot F$) not only drives the residual circulation, but it is also related to eddy mixing of potential vorticity $q$. Under the quasi-geostrophic (QG) approximation and neglecting parameterized (gravity) wave forcing, this can be formulated as

$$\nabla \cdot F = \overline{v'q'}, \tag{6}$$

where $v'$ and $q'$ denote the deviation of their means $\overline{v}$ and $\overline{q}$, respectively. Note that due to availability, we show the full

EPfd fields in Fig. 9 and not the QG EPfd. Note also, that strictly this relation holds true on the beta plane (Andrews et al., 1987), or approximately on isentropic surfaces, i.e. on $\Theta$ levels, on which mixing takes place. Therefore, the quantity may be somewhat distorted on the pressure levels presented here. This may lead to differences in the quantities, however, not qualitatively different conclusions in the following. Commonly, a flux-gradient relationship is assumed for the eddy PV flux, so that $\overline{v'q'} = -K_{yy}\partial q_y$, with the diffusivity coefficient $K_{yy}$. $\partial q_y$ is the gradient of the QG PV (see e.g. Edmon Jr. et al., 1980;

Cohen et al., 2014) with

$$\partial q_y = \frac{\partial \overline{q}}{\partial y} = \beta - \frac{\partial^2 \overline{u}}{\partial y^2} + \frac{\partial \left( f \cdot \frac{\partial \overline{\Theta}}{\partial y} / \frac{\partial \overline{\Theta}_0}{\partial p} \right)}{\partial p}. \tag{7}$$





**Figure 9.** Multi-model mean differences ($\Delta$) of (a) the zonal wind $\overline{u}$, (b) the EP flux divergence, (c) the meridional residual circulation $v^*$, (d) the meridional PV-gradient ($\partial PV/\partial y$) (e) the diffusivity coefficient $K_{yy}$ and (f) the ratio $K_{yy}/|v^*|$ between the periods 1970-1990 and 2080-2100 of the 8 (see main text) CCMI REF-C2 simulations. The contour lines show the multi-model mean climatology of the first period of the respective quantity. Stippled regions show where the statistical significance of the difference is below the threshold 95%.



Similarly as in Abalos et al. (2017), we calculate the diffusivity coefficient $K_{yy}$ as

$$K_{yy} = \frac{-\nabla \cdot F}{\partial q_y}. \tag{8}$$

This relation states that horizontal eddy mixing is proportional to the EP flux divergence, and indirectly proportional to the meridional PV gradient. Thus, a weak PV gradient indicates strong mixing, while a strong PV gradient acts as barrier to mixing,
and thus mixing is suppressed.

The MMM differences show that next to the enhanced wave dissipation in most of the stratosphere ($-\nabla \cdot F$ increases), also the PV gradient increases in the subtropics to mid-latitudes (see Fig. 9d) due to the strengthened winds. Thus, $K_{yy}$ tends to be enhanced due to increased wave dissipation, but reduced due to the increased PV gradient. As seen in Fig. 9e, the diffusivity changes are dominated by the increase in wave dissipation in the lower stratosphere (between around 100 to 50 hPa
in both hemispheres) and dominated by the decrease in wave dissipation below. This means that the vertical shift of EP flux convergence is reflected in $\Delta K_{yy}$, or in other words, that the region of strong mixing is shifted upward and the absolute strength of mixing increases in the lower stratosphere. At higher altitudes, the PV gradient contributes more strongly to $K_{yy}$, and in the SH mid-latitudes, the mixing strength $K_{yy}$ even decreases despite enhanced wave dissipation due to the strongly enhanced PV gradient. In the NH, $\Delta K_{yy}$ is smaller and mostly positive in the MMM. However, it should be noted that particularly in the
lower stratospheric subtropics, the region of the climatological maxima of orographic gravity wave drag, the QG assumption can be misleading due to the neglection of parameterized wave drag.

The mixing efficiency derived from the AoA data represents the relative strength of mixing, i.e. the ratio of the mixing mass flux to the residual meridional mass flux. The mixing mass flux is proportional to the mixing velocity, that can be expressed as $\overline{v_{mix}} = K_{yy}/L$ for a given horizontal length scale L. Thus, the ratio of mixing versus residual mass flux can be approximated
as

$$\frac{\overline{v_{mix}}}{\overline{v^*}} = \frac{1}{L} \frac{K_{yy}}{\overline{v^*}}. \tag{9}$$

Given a constant length scale $L$, the mixing-to-mean-advection ratio is proportional to $K_{yy}/|\overline{v^*}|$. As shown above, residual transport as well as mixing increases in most regions (Fig. 9c and e). Fig. 9f, however, shows that the ratio $K_{yy}/|\overline{v^*}|$ decreases in large parts of the stratosphere. This means that mixing increases less strongly (or even decreases) than residual transport.
Only in the (sub-)tropical lower to mid stratosphere, the relative mixing strength increases. The changes in $K_{yy}/|\overline{v^*}|$ mostly reflect the inverse change in the PV gradient, because the mixing diffusivity is inversely related to it. Note that if assuming $\overline{v^*} = -f^{-1}\nabla \cdot F$, the ratio $K_{yy}/|\overline{v^*}|$ equals $f/\partial q_y$, i.e. the ratio is only related to the PV gradient. Changes in $f/\partial q_y$ are overall similar to changes in $K_{yy}/|\overline{v^*}|$ (not shown), except for some details that might be related to the simplification of the equations and/or to the neglection of parametrized wave drag (i.e. small scale gravity wave drag).
Overall, the analysis presented above suggest that enhanced wave dissipation (caused by the zonal mean wind increase, see Shepherd and McLandress, 2011) amplifies the residual circulation as well as mixing, particularly in the lower stratosphere. However, the zonal mean wind changes also cause changes in the meridional PV gradient, and the stronger PV gradients act to inhibit mixing. Therefore, mixing increases less strongly than residual transport does, which explains the decrease in the mixing efficiency in the future (see Sect. 3.2).





However, as also shown in Sect. 3.2, the mixing efficiency does not decrease in all models. In Fig. 10, the mean $\Delta K_{yy}/|\overline{v^*}|$ in the region of the subtropical barrier averaged between 20°S-40°S and between 20°N-40°N is shown with altitude for the subset of eight CCMI-1 model simulations.

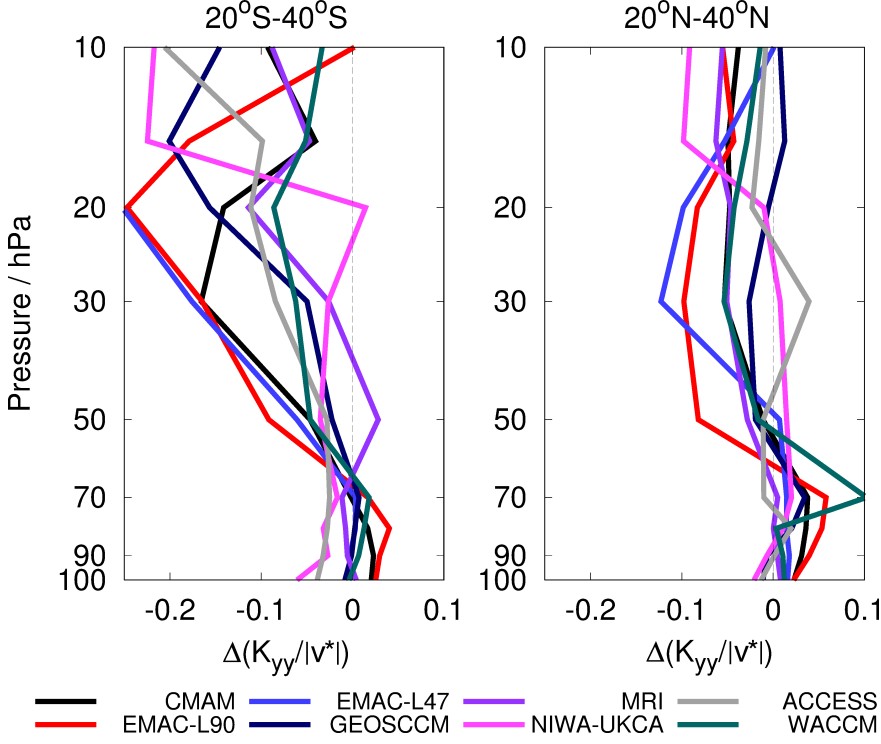

**Figure 10.** $\Delta(K_{yy}/|v^*|)$ (mean of 2080-2100 minus mean of 1970-1990) with altitude averaged between a) 20°S and b) 40°S as well as between 20°N and 40°N for eight of the CCMI model simulations.

$|\Delta(K_{yy}/|v^*|)|$ is larger in the SH as compared to the NH, in particular at higher altitudes. This is due to the acceleration of the Antarctic polar vortex (Fig. 9a) that leads to an increase in $\partial q_y$ (Fig. 9d), thereby reducing effective mixing. In most models, $\Delta(K_{yy}/|v^*|)$ is negative within these latitudes throughout the stratosphere in the SH as well as in the NH. Between 20 and 50 hPa there are only two exceptions, namely NIWA-UKCA and ACCESS in the NH, those two models that also have a slight positive $\Delta\epsilon$ (see Tab. 2). Moreover, the two EMAC model versions, which have the largest epsilon change also show the largest $|\Delta K_{yy}/|v^*||$, in both hemispheres. Although we can not find a clear inter-model correlation between $\Delta\epsilon$ and in $\Delta(K_{yy}/|v^*|)$ here, these are strong indications of a link between the two quantities.

Hence, the inter-model differences in mixing efficiency changes are consistent with differences in the relative rate of change in mixing to residual transport. The differences in $K_{yy}/|\overline{v^*}|$ changes between the models appear to be related to structural differences in zonal wind changes, i.e. the models with an increasing mixing efficiency (namely NIWA-UKCA, ACCESS) simulate zonal wind increases in the NH at higher latitudes than other models. In consequence, the PV gradient also increases





at higher latitudes, so that in the region of the subtropical barrier, mixing increases more strongly (see figures in Supplement). Our approximations as well as the limited number of models do not warrant a robust conclusion from inter-model correlations. However, the results shown here suggest that the mixing efficiency changes, as well as its inter-model spread are consistent with changes in the relative mixing strength due to changes in the background PV gradient.

## 5  4   Summary and conclusions

In the present study, we analyze the AoA differences of 10 CCMI-1 (Morgenstern et al., 2017) climate prediction simulations between the two periods 1970-1990 and 2080-2100. In agreement with previous model studies (Butchart et al., 2010; SPARC, 2010), we find a reduction of AoA over time in all model simulations. The smallest differences are consistently found in the tropics and the largest in the extratropical lower stratosphere, but the magnitude of the changes varies vastly among the models. Our investigation focuses on the reasons for this negative $\Delta$AoA (AoA(2080-2100)-AoA(1970-1990)) in the analyzed model simulations, as well as on the causes for the large model differences in $\Delta$AoA.

When we linearly separate $\Delta$AoA into the differences that are driven by aging by mixing (A_mix) changes and by residual circulation transport time (RCTTs) changes (Birner and Bönisch, 2011; Garny et al., 2014), we find that generally $\Delta$A_mix dominates in all models. In particular, the influence of $\Delta$A_mix controls almost the entire changes in AoA in the subtropical lower stratosphere. $\Delta$RCTT is important in the tropical pipe and in the downwelling branches of the extratropics, but only dominate in some models. This linear separation of $\Delta$AoA in its components, however, is intricate in its interpretation. First, A_mix itself is a function of (i.a.) the residual circulation and therefore the individual processes are not independent from each other and second, this dependence between the processes is not necessarily local. By means of inter-model correlation analyses, we find that the changes in RCTTs and AoA are locally correlated only in the extratropical middle stratosphere. The connection between the changes in tropical upwelling (at 70 hPa) and AoA, in contrast, are spread throughout the stratosphere, which also points towards a non-local dependence of AoA from residual transport. The changes of the residual circulation as a consequence of tropical upwelling changes had been associated mainly with a subtropical jet stream acceleration due to a thermal wind response to upper tropospheric warming and lower stratospheric cooling previously (see e.g. Lorenz and DeWeaver, 2007; Randel et al., 2008; Butchart, 2014). Recently, this was linked with the upward shift of the tropopause e.g. by Oberländer-Hayn et al. (2016) and Abalos et al. (2017). The variations of mixing over time and their impact on stratospheric circulation changes and thus AoA, however, are widely uncharted up to now.

In order to gain a deeper insight into the changes of mixing, we further investigate the mixing efficiency $\epsilon$ (Garny et al., 2014). This is a diagnostic quantity that is calculated by fitting the equations of a simplified model (the TLP model; Neu and Plumb, 1999) to the GCM data and provides information about the ratio of the mixing mass flux to the net mass flux. The mixing efficiency controls the strength of additional aging by mixing. Here, we reveal that the mixing efficiency decreases over time in most of the model simulations, but two models show a weak and one shows a strong positive $\Delta\epsilon$. A decrease in the mixing efficiency indicates that mixing does not increase as strongly as the residual circulation mass flux does. We find that in models with a stronger decrease in mixing efficiency, AoA increases more strongly, as less relative mixing reduces the





fraction of air that recirculates around the BDC branches (Garny et al., 2014). Hence, stronger tropical upwelling as well as a reduced mixing efficiency both lead to a decrease of AoA. The changes in mixing efficiency over time found here in the CCMI simulations contrast the results presented in Garny et al. (2014), who found a constant mixing efficiency in different climate states. However, as the changes in mixing efficiency appears to be model dependent, a zero change in mixing efficiency lies

within the range of $\Delta\epsilon$ found here. Moreover, note that the analysis of PV gradient changes presented in Garny et al. (2014) also suggest a decrease of the relative mixing strength (see their Fig. 11).

Subsequently, we quantify the influence of changes in mixing efficiency for $\Delta$AoA as well as the impact of $\epsilon$ variations on the $\Delta$AoA model spread for all model simulations, individually. The influence of $\Delta\epsilon$ on $\Delta$AoA is determined by means of calculating AoA for 2100 but with the $\epsilon$ of 1970 (AoA*) with the TLP model and then comparing the hypothetical $\Delta$AoA*

with $\Delta$AoA. We obtain a multi-model mean of 10.4% of the influence of mixing efficiency changes on the differences in AoA. However, the models show a large spread in this quantity ($\sigma$=18.3%) and as three models possess a positive $\Delta\epsilon$, also the sign is not consistent among the models. In a similar manner, we assess the impact of mixing efficiency variations on the model spread in $\Delta$AoA. This reveals that first, the AoA changes are generally smaller when $\epsilon$ is kept constant (at the 1970-1990 mean values) and second, that some of the model spread is caused by variations in $\epsilon$. This means that model differences in $\epsilon$ changes

lead to a considerable enhancement of the model inconsistencies in future projections of the BDC.

To study the reasons for the changes in the mixing efficiency, we analyze several dynamical fields as multi-model means of the differences between the two periods. These show the well-known coherence that the increase and upward shift of the zonal mean winds lead to enhanced wave dissipation in the lower stratosphere and thereby an amplified poleward residual transport. However, changes in wave dissipation also lead to variations in the properties of mixing. To see if this is reflected in the model

data, we calculate a diffusivity coefficient that bases on the ratio of wave driving to the meridional potential vorticity gradient (under the premise of QG theory). This reveals that the increase in wave dissipation shifts the region of strong mixing upward, thereby increasing the absolute strength of mixing in the lower stratosphere. Above, however, an enhanced PV gradient (which is due to the zonal wind increase) leads to a decrease in mixing strength. This counteraction of the two effects can explain why residual transport increases stronger than mixing and thus the negative $\Delta\epsilon$ in most of the models. In the models with an

increase in $\epsilon$, the relative mixing strength is consistently found to increase, in particular in the NH, because the zonal wind changes, and thus the PV gradient changes, take place at higher latitudes in these models. However, detailed process analysis experiments are required to test how robust this connection is.

Overall we separated the effects of mixing changes from changes in residual circulation in causing the decrease in AoA. We found that a decrease in the relative mixing strength leads to a further reduction of about 10% in AoA in the future in most

models. We further could show that the inter-model differences in simulated changes in the mixing efficiency contribute to the inter-model spread in the simulated AoA changes. The decrease in the relative mixing strength appears to be related to changes in background PV gradients. However, clear causalities can only be determined with model experiments that are specifically designed for that purpose, e.g. by varying certain parameters or model characteristics. The influence of mixing on the BDC and its changes can be important to project future climate conditions. We therefore suggest to conduct more in-depth analyses





with the aim to study the changes in residual circulation and mixing as well as their uncertainties and possible connections in more detail.

Supplementary material related to this article is available online at doi:10.5194/acp-0-1-2018-supplement.

*Author contributions.* RE, SD and HG designed and conducted the analysis and wrote the paper. PŠ, TB and HB helped with discussions on content and structure of the study. In their role as CCMI model PIs, the other authors contributed information concerning their individual models and helped revise the article.

*Data availability.* All CCMI-1 data used in this study can be obtained through the British Atmospheric Data Centre (BADC) archive (ftp://ftp.ceda.ac.uk, last access: May 2018). CESM1-WACCM data have been downloaded from http://www.earthsystemgrid.org (last access: May 2018). For instructions for access to both archives see http://blogs.reading.ac.uk/ccmi/badc-data-access.

*Competing interests.* The authors declare that they have no conflict of interest.

*Acknowledgements.* This study was funded by the Helmholtz Association under grant VH-NG-1014 (Helmholtz-Hochschul-Nachwuchs-forschergruppe MACClim). We thank the modelling groups for making their simulations available for this analysis, the SPARC/IGAC Chemistry-Climate Model Initiative (CCMI) project for organizing and coordinating the model data analysis activity and the British Atmospheric Data Center (BADC) for collecting and archiving the CCMI model output. ACCESS-CCM runs were supported by Australian Research Council's Centre of Excellence for Climate System Science (CE110001028), the Australian Government's National Computational Merit Allocation Scheme (q90) and Australian Antarctic science grant program (FoRCES 4012). We also acknowledge the project ESCiMo (Earth System Chemistry integrated Modelling), within which the EMAC simulations were conducted at the German Climate Computing Centre DKRZ through support from the Bundesministerium für Bildung und Forschung (BMBF). Moreover, we acknowledge the UK Met Office for use of the MetUM. This research was supported by the NZ Governments Strategic Science Investment Fund (SSIF) through the NIWA programme CACV. Olaf Morgenstern acknowledges funding by the New Zealand Royal Society Marsden Fund (grant 12-NIW-006). The authors wish to acknowledge the contribution of NeSI high-performance computing facilities to the results of this research. New Zealands national facilities are provided by the New Zealand eScience Infrastructure (NeSI) and funded jointly by NeSIs collaborator institutions and through the Ministry of Business, Innovation & Employments Research Infrastructure programme (https://www.nesi.org.nz). Petr Šácha was supported by the Government of Spain under grant no. CGL2015-71575-P and partly by GA CR under grant nos. 16-01562J and 18-01625S. Moreover, we thank Pooja Verma for useful comments on the manuscript and Andreas Engel for providing the in-situ AoA data.



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
