# Peer review of "The influence of mixing on stratospheric circulation changes in the 21st century"

_Atmospheric Chemistry and Physics, 2018_

## Referee Comment (RC1) · Anonymous Referee #1 · 20 Nov 2018

This manuscript presents an examination of the changes in the age of stratospheric air in CCMI model projections for the 21st century. This includes not only an analysis of the change in the age but also an analysis of the relative role of changes in mixing and residual circulation in these changes and an analysis of changes in the mixing efficiency. This analysis shows that changes in the mixing rather than just advection by the residual circulation plays a major in simulated decreases in the age of air. Further, differences in the mixing efficiency are related to differences in PV gradients. These are very important result, and greatly improve our understanding of modeled changes in stratospheric age of air.

This manuscript is certainly worth publishing in ACP, and it could be accepted as is. However, I think it would be great improved if the writing was made more concise and

if numerous grammatical errors were corrected. The specific comments below include some examples, but the need for more concise and grammatically correct writing applies throughout the paper.

SPECIFIC COMMENTS

1. I think the manuscript would be improved if it was more concise. This applies for the whole manuscript, but I think is highlighted by the final section. The final section should summarize the results of the study, and does not need to repeat information on the analysis steps. Nearly all paragraphs start with 1-2 sentences that are not needed, e.g.,

p 23, l 12-14 "When we linearly separate ... we find that generally ..." could be written simply as "Linear separation of the DeltaAoA into changes by mixing and residual circulation shows that the contribution due to mixing (DeltaA_mix) dominates."

p23, l 27-29 "In order to gain ..." These 2 sentences could be removed and just say "We have shown that mixing efficiency controls ..."

p24, l 7: First 2 sentences again not needed, or at least replaced by a short sentence. The method of separation does not need to be described in summary.

2. I think "age of air" rather than "circulation" should be in the title. The focus of the paper is on the age, and many people will read "circulation" as the residual mean circulation.

3. The order of first and last names of the authors need to reversed in the author list.

4. There are lots of grammatical errors. A few are listed below, but this is only a small subset. The paper needs a very careful proof reading.

pg 7, l 21: "They found out that ..." should be "They showed that ..."

p 12, l 6: "Moreover, as the only model, ULAQ"

pg 12, l 11: "reddish colours" refer to values and not colors.

p 12, l 23: "vstar integral". Should at least be "v*" but even then not a good description.

p 12, l 27: "Note that for this, data"

p 13, l 8: "have found out" See above.

p 13, l 11: "we now lay our focus"

p 14, l 15: "In consequence"
* * *

---

## Referee Comment (RC2) · Anonymous Referee #2 · 5 Dec 2018

The paper presents an analysis of the trends in the Brewer-Dobson circulation in 10 chemistry-climate models of CCMI-1. The contributions of residual circulation and mixing to the changes in AoA are separated by computing RCTT, A_mix, and mixing efficiency for each model. It is found that in most models the mixing efficiency decreases throughout the 21st century, and this explains about 10 % of the AoA changes. It is shown that different evolutions of mixing efficiency can partly explain the spread in the AoA model trends. Finally, it is argued that the decrease in mixing efficiency can be attributed to changes in the PV gradient, which increases due to stronger stratospheric westerly jets in the future.

The paper is very well suited for ACP and the CCMI special issue, provides novel results which make advances in understanding the future trends in the BDC and the role

of mixing in predicting the future evolution of stratospheric circulation. I do not have major comments on the content, but the paper would benefit from some work on the writing to clarify and simplify the message (especially in sections 3.2 and 3.3). I recommend publication after the following suggestions to improve the writing are addressed.

*Specific comments:*

- P1 L5 and P23 L9: could you provide a rough number to quantify the model spread in AoA trends? (also in the conclusions, on P23 L9)

- P3 L31-32: It is not clear what is the difference with the previous sentence.

- P6 L20-21: I do not understand why a term is needed to correct for the altitude dependence of the vertical residual circulation, when w* is already expressed as a function of z in Eq. (1). Perhaps you could briefly explain this?

- P6 L18: Is alpha a function of z? If so, it should be reflected in Eq. (1).

- P6 L29: Did you use wstar provided by the models or compute it from vstar?

- P6 L30: The tropical profiles provided for the TLP model: profiles of what variables?

- P7 L1: obtained by a best fit: of the TLP parameters to the model's age of air? (specify what is fitted)

- P7 L32 – P8 L2: Remove this sentence, already mentioned before.

- Figure 2: Perhaps it is worth mentioning that the observations have a much larger spread than the models' variability.

- P9 L13-14: (as both. . .) : this cannot be concluded from those papers

- P10 L3-4: This separates. . . : Not really separates, but you can argue that, based on the sensitivity experiments in Polvani et al. 2018 GRL showing that the change in the slope is due to ODS, one can capture the GHG effect alone by considering the net trends from 1970-2100.

- P12 L11: GEOSCCM does not belong to this group in my opinion, it is more similar to SOCOL and WACCM. Also, NIWA and ACCESS are very similar.

- P12 L16: 'slightly positive': remind here that this means values of the ratio above 1.

- Figure 4b: over which latitude band do you average w* at 70 hPa?

- P13 L14: '10 year moving average': this averaging is not mentioned in Section 2.2.

- P14 L3-4: in MRI and EMAC-L90 the trends are near zero in the first period, the only model that shows a clear increase in epsilon in the first period is ULAQ.

- Figures 4 and 6: These correlations are based on ten points (one for each model). It would be interesting to see, perhaps not for the paper, what do the correlations look like for one single model on interannual variability. Do they show similar features?

- P16 L4: it would be clearer for the reader if the intermediate step AOA'(2100)=A_mix(1970)+RCTT(2100) were included.

- P17 L9-13: This description of Table 3 is confusing. The sign of the fractional impact of mixing efficiency on AoA changes is negative for three models. It seems to me that this should be the first thing mentioned and explained clearly. The models in which this contribution is negative are the same only three models for which $\Delta\epsilon$ is positive. So in these models the mixing efficiency is increasing over

time, and that is why they make an opposite contribution to AoA trends. Is this correct?

- P18 L7-8: However, it is not strictly coincidental... I do not understand this sentence.

- P23 L3: AoA decreases

*Technical:*

- P3 L23: aging by mixing, and residual. ...

- P6 L10: via → from

- P6 L16: add a reference (Neu and Plumb 1999, Garny et al. 2014)

- P6 L26: that counts → valid

- P6 L27-28: remove 'however' or 'nevertheless'

- P7 L2: remove 'therefore'

- P7 L5: indirectly → inversely

- P7 L20-22: Is this paper published? Otherwise it should not be cited. Also on P12 L4-5.

- P7 L23: remove 'it were'

- P7 L24: remove 'and these models'

- P7 L26: the lower end
- Table 2 caption: typo 1980 → 2080

- P10 L16: Hence → Specifically

- P14 L6: filter out

- P14 L13-14: Rephrase: The mechanisms for the mixing changes are diagnosed using the potential vorticity gradient in Section 3.4.

- P15 L3: remove ','

- P15 L7: remove 'also'

- P15 L8-9: Rephrase: connected with changes in both mixing and residual circulation.

- P16 L10: remove the second 'difference'.

- P17 L5: Remove the first sentence (repetitive), start with: The difference between...

- P18 L4: model range → model spread

- P18 L4: (from 0.35 to 0.22) units? (also missing on Fig. 8).

- P18 L9: remove 25%-33%

- P19 L3: '...mean (MMM) diagnostics' (without the final s)

- P19 L24: 'zonal means', 'due to data availability'

- P19 L29: remove Jr.

- P21 L1: (2016) Also correct the reference, the 2017 JAS paper is different from the 2016 JAS paper (both cited here).

- P21 L3: indirectly → inversely

- P22 L1: remove 'also'

- P22 L9: remove 'in'

- L10: cannot

- P23 L17: remove (i.a.)

- P24 L4: appear

- P24 L9: the obtained hypothetical. . .

- P24 L24: increases faster and this $\Delta\epsilon$ is negative. . .

- P24 L28: Overall → In summary

- P24 L29: remove 'a further'

- P24 L30: we further showed

---

## Author Comment (AC1) · 21 Dec 2018

Dear anonymous referee #1,

thank you for investing time and effort to review the paper.

Please find below our point by point answers (in black) to your comments (in blue).

Best regards

Roland Eichinger, Simone Dietmüller and Hella Garny

[Figure]

This manuscript presents an examination of the changes in the age of stratospheric air in CCMI model projections for the 21st century. This includes not only an analysis of the change in the age but also an analysis of the relative role of changes in mixing and residual circulation in these changes and an analysis of changes in the mixing efficiency. This analysis shows that changes in the mixing rather than just advection by the residual circulation plays a major in simulated decreases in the age of air. Further, differences in the mixing efficiency are related to differences in PV gradients. These are very important result, and greatly improve our understanding of modeled changes in stratospheric age of air.

This manuscript is certainly worth publishing in ACP, and it could be accepted as is. However, I think it would be great improved if the writing was made more concise and if numerous grammatical errors were corrected. The specific comments below include some examples, but the need for more concise and grammatically correct writing applies throughout the paper.

We have corrected the errors you found, those of reviewer #2 and also went through the paper again and found and corrected a few more. We are convinced that proof reading will capture the remainders. The same accounts for conciseness of the article, but see also the marked-up "diff" file that will be attached to the final resonse.

I think the manuscript would be improved if it was more concise. This applies for the whole manuscript, but I think is highlighted by the final section. The final section should summarize the results of the study, and does not need to repeat information on the analysis steps. Nearly all paragraphs start with 1-2 sentences that are not needed, e.g.,

See above, we have made efforts to make the paper more concise.

p 23, l 12-14 When we linearly separate ... we find that generally ... " could be written

simply as "Linear separation of the DeltaAoA into changes by mixing and residual circulation shows that the contribution due to mixing (DeltaA_mix) dominates."

Thanks, done.

p23, l 27-29 "In order to gain ... " These 2 sentences could be removed and just say "We have shown that mixing efficiency controls ..."

Done.

p24, l 7: First 2 sentences again not needed, or at least replaced by a short sentence. The method of separation does not need to be described in summary. 2. I think "age of air" rather than "circulation" should be in the title. The focus of the paper is on the age, and many people will read "circulation" as the residual mean circulation.

Done.

3. The order of first and last names of the authors need to reversed in the author list.

Done.

4. There are lots of grammatical errors. A few are listed below, but this is only a small subset. The paper needs a very careful proof reading.

See above, we have corrected many mistakes now, proof reading will certainly find the remainders.

pg 7, l 21: "They found out that ... " should be "They showed that ... "

Done.

[Figure]

p 12, l 6: "Moreover, as the only model, ULAQ"

Changed to read:

*Moreover, ULAQ is the only model that ...*

pg 12, l 11: "reddish colours" refer to values and not colors.

Changed to

*regions where the differences of residual transport accounts for less than 50% of △AoA*

p 12, l 23: "vstar integral". Should at least be "v*" but even then not a good description.

Done.

p 12, l 27: "Note that for this, data"

Done.

p 13, l 8: "have found out" See above.

Done.

p 13, l 11: "we now lay our focus"

Changed to

*we now focus*

p 14, l 15: "In consequence"

Done.

---

## Author Comment (AC2) · 21 Dec 2018

Dear anonymous referee #2,

thank you for investing time and effort to review the paper.

Please find below our point by point answers to your comments.

Best regards

Roland Eichinger, Simone Dietmüller and Hella Garny

[Figure]

The paper presents an analysis of the trends in the Brewer-Dobson circulation in 10 chemistry-climate models of CCMI-1. The contributions of residual circulation and mixing to the changes in AoA are separated by computing RCTT, A_mix, and mixing efficiency for each model. It is found that in most models the mixing efficiency decreases throughout the 21st century, and this explains about 10% of the AoA changes. It is shown that different evolutions of mixing efficiency can partly explain the spread in the AoA model trends. Finally, it is argued that the decrease in mixing efficiency can be attributed to changes in the PV gradient, which increases due to stronger stratospheric westerly jets in the future. The paper is very well suited for ACP and the CCMI special issue, provides novel results which make advances in understanding the future trends in the BDC and the role of mixing in predicting the future evolution of stratospheric circulation. I do not have major comments on the content, but the paper would benefit from some work on the writing to clarify and simplify the message (especially in sections 3.2 and 3.3). I recommend publication after the following suggestions to improve the writing are addressed.

Thanks, we appreciate your efforts providing suggestions to improve the paper. We have taken them into account and additionally corrected some more errors. Please see below and also the marked-up "diff" file that will be attached to the final resonse.

P1 L5 and P23 L9: could you provide a rough number to quantify the model spread in AoA trends? (also in the conclusions, on P23 L9)

$\sigma(\Delta AoA) = 0.18$ for $\mu(\Delta AoA) = 0.54$
We included this in the results and in the conclusions section, but, for the sake of readability, refrained from including it in the abstract.

P3 L31-32: It is not clear what is the difference with the previous sentence.

The difference is that one tackles climatologies, one differences between two climate states. This is now more clear, the sentences now read:

*In the companion paper, Dietmueller et al. (2018) have already shown that the mixing efficiency can explain most of the AoA model spread in the climatologies from 1960 to 2010. In the present study, we quantify the impact of mixing efficiency (relative mixing strength) differences between two climate states in the individual model simulations.*

P6 L20-21: I do not understand why a term is needed to correct for the altitude dependence of the vertical residual circulation, when w* is already expressed as a function of z in Eq. (1). Perhaps you could briefly explain this?

That is basically just the analytical solution of the TLP equation for an altitude-dependent w*. The term comes from the horizontal advection between z and $z_T$, so $M(z)w(z) - M(z_T)w(z_T)$.... However, as this is not really descriptive as an explanation, we will simply add:

*(additional analytical solution term from horizontal advection, for details see Neu and Plumb (1999) and Garny (2014))*

P6 L18: Is alpha a function of z? If so, it should be reflected in Eq. (1).

Done.

[Figure]

**P6 L29: Did you use wstar provided by the models or compute it from vstar?**

For consistency between the models, we also used v* here. We added this to the text. Please also refer to the Supplement of the companion paper Dietmüller et al. (2018) for this topic.

**P6 L30: The tropical profiles provided for the TLP model: profiles of what variables?**

$\overline{w}^*$, tropopause height and AoA. Included in the text now.

**P7 L1: obtained by a best fit: of the TLP parameters to the model's age of air? (specify what is fitted)**

Only epsilon is fitted. AoA and w* are the input and one calculates a best fit of these profiles for the TLP equations via the optimization. We have changed the sentece to read:

*The mixing efficiency is then obtained by the TLP model's best fit to the CCM AoA profile. Here, the best fit is done for the altitude range from the tropopause to [32]km (details for the calculation of the mixing efficiency are given in Garny et al. (2014)).*

**P7 L32 – P8 L2: Remove this sentence, already mentioned before.**

Done.

Figure 2: Perhaps it is worth mentioning that the observations have a much larger spread than the models' variability.

This is indeed an interesting detail, but we do not want to include it here. This is already the first step into the analysis of the observations in comparison with models. Other studies have done it before and this would open up a can of worms which we intended to leave closed as for this paper.

P9 L13-14: (as both ... ) : this cannot be concluded from those papers

We have restructured the sentence to separate clearly which statement refers to the citations and which does not, it now reads:

*Several studies (citations) have recently pointed out the importance of the role of ODSs for the trends in stratospheric dynamics. ODSs act as both, radiatively active greenhouse gases and chemically active gases controlling ozone depletion and recovery.*

P10 L3-4: This separates ... : Not really separates, but you can argue that, based on the sensitivity experiments in Polvani et al. 2018 GRL showing that the change in the slope is due to ODS, one can capture the GHG effect alone by considering the net trends from 1970-2100.

Thanks, that is helpful. We changed the sentence to:

*Based on the sensitivity simulations in Polvani et al. (2018), which shows that the change in the slope in the year 2000 is due to ODSs, this allows to capture the GHG effect alone.*

P12 L11: GEOSCCM does not belong to this group in my opinion, it is more similar to SOCOL and WACCM. Also, NIWA and ACCESS are very similar.

Right! Thanks for checking carefully. The groups are now

(EMACL47, EMACL90, GEOSCCM, SOCOLv3)

and

(GEOSCCM, ACCESS, NIWA-UKCA, WACCM)

P12 L16: 'slightly positive': remind here that this means values of the ratio above 1.

Added:

*(i.e. $\Delta AoA/\Delta RCTT>1$)*

Figure 4b: over which latitude band do you average w* at 70 hPa?

Over the individual turn-around latitudes. We included that in the caption now.

P13 L14: '10 year moving average': this averaging is not mentioned in Section 2.2.

We included the sentence:

*For analysis, we calculate the ten year running averages of $\epsilon$ to obtain climatologically representative values.*

P14 L3-4: in MRI and EMAC-L90 the trends are near zero in the first period, the only model that shows a clear increase in epsilon in the first period is ULAQ. Figures 4 and 6: These correlations are based on ten points (one for each model). It would be interesting to see, perhaps not for the paper, what do the correlations look like for one single model on interannual variability. Do they show similar features?

Corrected, the paragraph now reads:

*For example in MRI and in EMAC-L90, $\epsilon$ first is almost constant and then it decreases and in GEOSCCM and CMAM, $\epsilon$ first decreases and in the later period it rises. Only the ULAQ model shows a positive trend in both periods.*

P16 L4: it would be clearer for the reader if the intermediate step $AOA'(2100) = A\_mix(1970) + RCTT(2100)$ were included.

Done.

P17 L9-13: This description of Table 3 is confusing. The sign of the fractional impact of mixing efficiency on AoA changes is negative for three models. It seems to me that this should be the first thing mentioned and explained clearly. The models in which this contribution is negative are the same only three models for which $\Delta\epsilon$ is positive. So in these models the mixing efficiency is increasing over time, and that is why they make an opposite contribution to AoA trends. Is this correct?

Thanks for pointing this out. To make the point more prominent, we have restructured the paragraph, it now reads:

*NIWA-UKCA, ACCESS and ULAQ show a negative fractional impact of mixing efficiency on AoA changes. These are the three models that also show a positive $\Delta\epsilon$ (see Tab. 2). In contrast to the other models, the mixing efficiency therefore leads to an AoA increase over time. The negative $\Delta RCTT$ therefore accounts for more than the entire negative $\Delta AoA$ to compensate the effect of the $\epsilon$ change. With less than 5.5% and 3.5%, NIWA-UKCA and ACCESS have the lowest contribution of $\Delta\epsilon$ on the AoA change and with up to 29%, ULAQ and EMAC-L90 have the largest.*

P18 L7-8: However, it is not strictly coincidental ... I do not understand this sentence.

Removed and rewritten, the part now reads:

*The model range decreases here because being the model with the largest $\Delta AoA$, EMAC-L90 has a negative $\Delta\epsilon$ and ULAQ, the model with the lowest $\Delta AoA$ has a positive $\Delta\epsilon$. A large/small (negative) $\Delta\epsilon$ causes a large/small $\Delta AoA$.*

P23 L3: AoA decreases

Done. (Meant was L8)

Technical:

We have also corrected all technical issues the referee mentioned